# Impaired Activity of Daily Living Status of the Older Adults and Its Influencing Factors: A Cross-Sectional Study

**DOI:** 10.3390/ijerph192315607

**Published:** 2022-11-24

**Authors:** Jin Gao, Qing Gao, Liting Huo, Jianchuang Yang

**Affiliations:** 1School of Humanities and Laws, Northeastern University, Shenyang 110169, China; 2School of Government, Sun Yat-sen University, Guangzhou 510275, China

**Keywords:** activities of daily living, older adults, chronic diseases

## Abstract

This study aimed to explore the impaired activity of the daily living ability status and its influencing factors among older adults. A sample of 10,148 participants (≥60 years old) who met the requirements for the activity of daily living scale was used in this study, and the Health and Aging Tracking Survey was selected for data collection. The impaired activities of the daily living status of older adults and their influencing factors were analyzed by single-factor descriptive analysis and multi-factor logistic regression. The study results showed that the rate of impaired activities of the daily living ability of participants was 26.56%, among which the rate of mild impairment was 17.34% and severe impairment was 9.22%. Multi-factor binary logistic regression analysis results showed that demographic characteristics, lifestyle habits, and physical health status were associated with older adults’ daily living activity ability. Among them, ages ≥80 years, male, habitual smoking, physical disability, and chronic diseases had a more significant impact.

## 1. Introduction

With the emergence of an aging society and the increasing progress of health care concepts, the ability of older adults to perform activities of daily living (ADL) has been recognized and recommended by WHO for geriatric epidemiological studies as an important criterion for assessing and studying their physical health status [1]. The ADLs are a series of basic activities that people must perform in their daily lives to take care of their clothing, food, housing and transportation, maintain personal hygiene, and live independently in the community. These activities are an important indicator of older adults’ health status. Related studies have shown that individual socioeconomic factors, somatic health, depression status, and health insurance influence ADL function [2]. The ability to perform ADLs is also the most important indicator to evaluate the independent living ability of older adults [3]. WHO pointed out that the leading indicators of their health evaluation should be not only death and disease indicators but also the ability to live independently [4]. Older adults who smoke and drink too much alcohol have a low capacity for ADLs; while engagement in physical work, recreational activities, and exercise can moderate the effects of negative lifestyles, resulting in a higher capacity [5,6]. Therefore, the study of the impaired ADL of older adults and their influencing factors can help to fully understand the mechanism of health determination and the path of intervention, assist in formulating scientific health interventions, improve the living conditions of older adults, and build a more efficient economic support system for the older people.

Scholars have studied the impaired ADLs of older adults and their influencing factors in two main ways: First, they have used different research methods. For example, some scholars have conducted a comprehensive assessment of older adults over 65 and analyzed the relationship between muscle strength, cognitive function, ADLs, and depression variables in an experiment in which statistical significance was found in all variables [7]. Others have analyzed the mediating effect of ADL ability between diabetes and depression in older adults using difference tests, stepwise regression analysis, and bootstrap mediation tests [8]. A non-randomized controlled intervention trial was also used to divide older adults into a rehabilitation group and a non-rehabilitation group and to measure their ability to perform ADLs through continuous observation and assessment of the quality of life using the Philadelphia Geriatric Center Psychiatric Inventory [9]. Based on relevant literature, most studies on older adults’ ADL ability have used statistical analysis of quantitative studies as a tool, relatively mature ADL scales for scoring [10].

Second, they have studied the factors influencing the ability of older adults to perform ADLs. Different genders, the presence of chronic diseases, disabilities, and social activities were considered the main influencing factors on the ability to perform ADLs, as well as self-rated health, and depression in older adults [11]. Research to understand the relationship between ADLs and depressive symptoms [12] was undertaken. Some scholars have explored the relationship between ADLs, cognitive functioning, social support, and their own attitudes toward aging using the 2014 Chinese Longitudinal Survey on Aging Society and developed structural equation models to examine mediating effects. Other scholars have investigated physical functioning related to ADLs, considering the role of cognitive, psychological, and social factors and using physical, cognitive, psychological, and social instruments to investigate the basic activity (BADL) and instrumental activity (IADL) levels [13]. The analyses revealed that physical functioning was the only individual factor significantly associated with BADL and IADL levels in older adults [14]. Some scholars have found that depression has a significant impact on the ability of the elderly to perform activities of daily living, and the higher the level of depression, the more severely the elderly are impaired in activities of daily living [15]. Therefore, communities and families should take measures to screen and intervene in the mental health of the elderly in a timely manner to improve the ability of the elderly to perform activities of daily living. [16].

Based on the above scholarly literature, this paper analyzes the status of impaired ADLs in older adults and the factors affecting them. The baseline survey data from the 2015 China Health and Retirement Longitudinal Study (CHARLS) conducted in 28 provinces nationwide were used to analyze the degree of ADL impairment and its influencing factors among older adults aged 60 years and above and to provide constructive references for promoting their physical health and improving self-care ability.

## 2. Materials and Methods

### 2.1. Sample Selection and Representativeness

This study is based on data from the 2015 CHARLS, a baseline survey conducted by the National Development Research Institute of Peking University (NDRI) to collect high-quality microdata representing households and individuals aged 45 and older to analyze population aging in China. The survey was conducted in 150 counties and 450 communities (villages) in 28 provinces. In the summer of 2014, CHARLS conducted a life history survey of all respondents and provided a historical overview of 20,547 respondents out of a sample of 12,250 in the areas of family, marriage, childbirth, employment, education, migration, and health since their birth. In the summer of 2015, the second follow-up interview of the regular survey of the national baseline sample was conducted, and a total of 11,797 and 20,284 myriads were interviewed, with a follow-up success rate of 87%. A total of 412 village households were interviewed regarding their economic history, and 3797 individuals were interviewed about their family histories [17].

In terms of sampling method, CHARLS adopted strict random sampling, with countrywide county-level units randomly selected according to the PPS (probability proportional to size) method, stratified by region, urban and rural areas, and the average GDP of the people. Concurrently, in each stage of the sampling process, to avoid artificial manipulation, the project staff conducted the sampling using a computer program, and no substitution of samples was allowed. CHARLS successfully conducted regular follow-up visits to these baseline samples in 2013, 2014, and 2015, based on the 2011–2012 national large-scale baseline survey, to maintain the representativeness of the middle-aged population.

Based on the second follow-up interview of this survey and the supplemental survey in 2016, this study also selected older adults aged 60 years and above as the study subjects according to the study needs, screened out missing and outlier samples, and chose respondents who had answered the questions related to the ADL scale. Finally, a total of 10,148 samples were included in this study.

### 2.2. ADL

Based on the ADL scale, the components of the ADL abilities of the older adults were selected from the CHARLS data: Personal activities of daily living (PADL) abilities such as toileting, eating, dressing, walking, and bathing and IADL behavioral abilities such as telephone use, shopping, meal preparation, household chores, medication taking, and self-care economy [18].In this study, the variables were selected based on a review of the relevant literature. (1) Relevant factors that may affect the ability of older adults to perform ADLs were included as independent variables (e.g., demographic characteristics [19], lifestyle habits [20], health status [21], and other relevant factors). Data on gender, age, marital status, education level, exercise, smoking, drinking, disability, chronic diseases, and self-rated health status were used in the analysis to examine the effects on the ADL ability of older adults [22]. (2) Dependent variable: Degree of ADL impairment, which is a categorical rating variable assigned according to the scale scoring, with no impairment assigned as 0 and a joint assignment of mild impairment and severe impairment as 1, constituting an event with probability 1 (no impairment 0, impairment 1), i.e., a dichotomous variable. In this study, the ADLs of the respondents were scored according to data in the CHARLS database with a scale of 1 to 4 for each item, with scores of 1, 2, 3, and 4 representing what they could completely do, had some difficulty doing, needed help, and were entirely dependent on others to do, respectively, with a total score of 14 to 56. The higher the score, the higher the impairment in ADLs; a total ADL score of 14 indicated no impairment in ADL, 15–21 indicated mild impairment, and ≥22 indicated severe impairment.

### 2.3. Statistical Analysis

STATA.MP 17.0 and Excel performed the data collation phase to build the initial database and to filter, classify, and integrate the indicators and variables included in this study. The data analysis phase used SPSS 23.0 to build the final database, and single-factor descriptive statistics analysis and multi-factor binary logistic regression analysis were performed on the integrated and complete data.

## 3. Results

### 3.1. ADL Status

Among 10,148 elderly people aged ≥60 years, 5218 (51.42%) were aged 60–69 years, 3369 (33.20%) were aged 70–79 years, and 1561 (15.38%) were aged ≥80 years. The gender was male 4920 (48.48%) and female 5228 (51.52%). Furthermore, the sample of elderly people contained 5395 (53.16%) who smoked, 4753 (46.84%) who did not smoke; 2841 (28.00%) who had physical disabilities, 7307 (72.00%) who did not have disabilities; 2908 (32.37%) who had chronic diseases, 7240 (67.63%) who did not have chronic diseases. In the evaluation of their own health, 3865 people (38.09%) rated themselves as having good health, 5074 people (50%) rated themselves as having average health, and 1209 people (11.91%) rated themselves as having bad health. In addition, statistics were also collected on the marital status, education level, participation in exercise, and alcohol consumption of the elderly.

### 3.2. Impaired ADLs

Among the 11 ADL abilities, 7453 older adults had no impairment in ADLs. That is, 73.44% were not impaired in ADLs, and 26.56% were impaired (including 17.34% of mild impairment and 9.22% of severe impairment). Among those impaired, the mild and severe impairment rate was 23.72% and 12.66% for males and 10.31% and 5.97% for females, respectively. The mild and severe impairment rate was 0.54% and 0.25% for older adults aged 60–69 years, 31.97% and 4.99% for those aged 70–79, and 41.96% and 48.30% for those aged ≥80 years, respectively. By the end of 2015, the number of people over 60 years in China had reached 220 million. After the baseline survey in 2011–12, CHARLS calculated the sample weight according to the sampling procedure. The weighted CHARLS characteristics were often close to those in the 2010 census, indicating that these data fully represented middle-aged people in China [23].

### 3.3. Univariate Analysis of the ADL Ability of Older Adults (Table 1 and Table 2)

Comparing the ADL ability of older adults in different populations, the differences in gender, age, marital status, education, exercise, smoking status, alcohol consumption, whether disabled, whether suffering from chronic diseases, and self-rated health were statistically significant (*p* < 0.01).

**Table 1 ijerph-19-15607-t001:** Comparison of activities of daily living ability among older adults with different characteristics.

Demographic Characteristics	Category	Survey Number of People	Percentage (%)	Mildly Damaged Number of People	Mild Impairment Rate (%)	Severe Damage Number of People	Severe Impairment Rate (%)
Gender	Male	4920	48.48	1167	23.72 ^a^	623	12.66 ^a^
	Female	5228	51.52	539	10.31	312	5.97
Age (years)	60–69	5218	51.42	28	0.54 ^a^	13	0.25 ^a^
	70–79	3369	33.20	1077	31.97	168	4.99
	≧80	1561	15.38	655	41.96	754	48.30
Marital Status	Married	7254	71.48	1006	13.89 ^a^	106	1.46 ^a^
	Divorce	673	6.63	127	18.87	65	9.66
	Bereaved Spouse	2151	21.20	613	28.50	757	35.19
	Unmarried	70	0.69	14	20.00	7	10.00
Education level	Illiterate	5460	53.80	1262	23.11 ^a^	648	11.87 ^a^
	Primary School	3706	36.52	464	12.52	271	7.31
	Junior High School and above	982	9.68	34	3.46	16	1.63
Exercise	Exercise	5767	56.83	695	12.05 ^a^	181	3.14 ^a^
	No Exercise	4381	43.17	1065	24.31	754	17.21
Smoking status	Smoking	5395	53.16	1452	26.91 ^a^	902	16.72 ^a^
	No Smoking	4753	46.84	308	6.48	33	0.69
Drinking situation	Drinking	4545	44.79	1090	23.98 ^a^	580	12.76 ^a^
	No Alcohol	5603	55.21	670	11.96	355	6.34
Is disabled	Disability	2841	28.00	795	27.98 ^a^	789	27.77 ^a^
	No Disability	7307	72.00	965	13.21	146	2.00
Have a chronic disease	Yes	2908	32.37	1065	36.62 ^a^	934	32.12 ^a^
	No	7240	67.63	695	9.60	1	0.01
Self-assessment of health status	Not Good	1209	11.91	169	13.98 ^a^	566	46.82 ^a^
	General	5074	50.00	1522	30.00	362	7.13
	Good	3865	38.09	69	1.79	43	1.11

Note: ^a^: *p* < 0.01.

The results of the two-factor correlation analysis of all explanatory variables in the database showed that the correlation coefficients between the variables were less than 0.75, so there was no multicollinearity between the variables. After the Monte Carlo test was selected for the chi-square test data statistics, the conclusion reached was consistent with the results of Pearson’s correlation analysis (Appendix A).

**Table 2 ijerph-19-15607-t002:** Chi-square test.

	Value	Degree of Freedom	Progressive Significance (Bilateral)	Monte Carlo Significance (Bilateral)			Monte Carlo Significance (One-Sided)		
				Significance	99% Confidence Interval		Significance	99% Confidence Interval	
					Lower Limit	Upper Limit		Upper Limit	Upper Limit
Pearson Cardinal	472.698	2	0.000	0.000 ^b^	0.000	0.000	0.000 ^b^	0.000	0.000
likelihood ratio	478.551	2	0.000	0.000 ^b^	0.000	0.000			
Fisher Precision Test	478 299			0.000 ^b^	0.000	0.000			
Linear correlation	406.398	1	0.000	0.000 ^b^	0.000	0.000			
Number of active cases	10148								

Note: ^b^: *p* < 0.01.

### 3.4. Multi-Factor Analysis of the ADL Ability of Older Adults (Table 3)

Based on univariate analysis, the factors with significant correlation were selected as independent variables; the degree of impairment of ADLs of the older adults as dependent variables (0 = no impairment, 1 = impairment); and eight factors, including gender, age, marital status, education level, smoking, drinking, disability, and chronic diseases, as independent variables. The inclusion standard was 0.05, and the exclusion standard was 0.1. The regression results showed that older adults who are older (e.g., >70), male, smoking, disabled, and suffering from chronic diseases had greater ADL damage and that age is the risk factor for older adults’ high ADL damage rate. Furthermore, smoking factors were stratified for older adults according to years of smoke exposure (<20, 20–40, and 40+ years) and the average number of cigarettes smoked per day (<20, 20–40, and 40+ cigarettes). The results showed that the years of smoking (*p* < 0.01) and average number of cigarettes smoked per day (20–40 [*p* < 0.01] and above 40 [*p* < 0.05]) were associated with a higher rate of impaired ADL in older adults, indicating that the number of years of smoking (20–40) and the number of cigarettes smoked (>20) had a significant effect on older adults’ impaired ADL.

**Table 3 ijerph-19-15607-t003:** Multi-factor binary logistic regression analysis of factors influencing ADL in older adults.

Factors			Dummy Variable	*β*	*Sx*	Wald *χ*^2^ Value	*p*-Value	*OR* Value	*95% CI*
Gender		Female	0						
		Male	1	−0.929	0.277	11.254	0.001	0.395	0.230 to 0.680
Age		60–69	0						
		70–79	1	0.482	0.015	1093.425	0.000	1.196	1.071 to 1.334
		≥80	2	0.421	0.010	1821.358	0.000	1.524	1.495 to 1.554
Smoking Habit	Years of Smoking	<20	0						
		20-40	1	−1.097	0.194	32.001	0.000	0.334	0.228 to 0.488
		>40	2	0.018	0.094	0.037	0.848	1.018	0.847 to 1.224
	Average number of cigarettes smoked per day	<20	0						
		20–40	1	−0.314	0.093	11.519	0.001	0.731	0.609 to 0.876
		>40	2	−0.401	0.184	4.772	0.029	0.670	0.467 to 0.960
Is disabled		No	0						
		Yes	1	−0.779	0.117	44.585	0.000	0.459	0.365 to 0.577
Have a chronic disease		No	0						
		Yes	1	−3.140	0.132	568.608	0.000	0.043	0.033 to 0.056

## 4. Discussion

Older adults’ ADL ability is the key to their quality of life [24]. This study showed that the ADL impairment rate of Chinese older adults was 26.56%, which was higher than the 19.4% in Beijing [25] and 21.4% in Guiyang City, Guizhou Province [26], and lower than the 34.49% in Yichang Yiling Mountains [27]. The CHARLS sample was more representative; ADL impairment rate should be representative of the overall situation in China.

The results of this study showed that the rate of ADL impairment was higher in male than female older adults. This differs from Zhang’s (2005) study conducted in Anhui Province, China, which identified a higher prevalence of chronic diseases, cognitive decline, and a higher burden of family and daily caregiving among women as the main factors contributing to the higher incidence of ADL disability in female older adults [28]. This higher prevalence of functional disability was attributed to increased impairment and illness with age [29]. This is because gender is a crucial factor affecting the ability of older adults to perform ADL. Men, as the main manual workers in families, bear more significant family burdens and social responsibilities and, thus, suffer more physical and mental injuries than female older adults; they have poorer living habits than women and more social responsibilities because of their work. Thus, they risk physical injuries, and there is a potential increase in the threat of diseases, resulting in shorter life expectancy and a poorer ability to perform ADL and lead an active life when compared to female older adults [30].

Age also significantly impacts older adults’ ADL ability. The study results showed that ADL impairment increased with age, and somatic and IADL were severely impaired. This finding is consistent with that of Jiang et al. [31], who found that age is an important indicator of impaired ADL. The reason for this phenomenon is that as older adults enter their senior years, their body functions become impaired, their resistance is poorer, and the possibility of disease increases greatly; serious physical impairment leads to the gradual loss of basic abilities such as going to the toilet, walking, and bathing, which can be accomplished only with the help of others and can cause psychological and mental damage, depression [32], grumpiness, and loss of enjoyment of life. These changes can affect the performance of social functions such as shopping and financial management, thus, resulting in the overall impairment of ADL. The government and the community should focus on vulnerable older adults and formulate policies to improve their life activities. Family members should strengthen communication with them, provide them emotional comfort and life care, meet their psychological needs, and help prevent and control diseases [33,34,35].

Smoking can cause serious ADL impairment in older adults because excessive smoking increases the risk of cardiovascular disease, lung disease, chronic disease, and in severe cases, disability and even inability to care for themselves. Smoking can cause a simultaneous decrease in pulmonary ventilation and reserve function and an increase in airway resistance in respiratory asymptomatic individuals, and long-term smoking can seriously affect lung function [36]. Therefore, older adults should strengthen their health self-management, develop an active and healthy lifestyle and good lifestyle habits—such as a proper diet, quitting smoking and limiting alcohol, exercising regularly, developing an optimistic health mindset, and strengthening their contact with the outside world—to promote the improvement of ADL.

Disability and chronic diseases are important factors affecting older adults’ ability to perform ADL [37]. Those with disabilities are less able to take care of themselves, and older adults with chronic diseases are limited in their ability to exercise, and their ability to express themselves is also greatly reduced [38,39]. Ćwirlej-Sozańska et al. proposed the development of a scoring system based on a ratio scale for measuring disability in personal and social ADL to establish the degree of disability in older adults [40]. Older adults with chronic diseases have reduced physical functions, weak resistance, and poor mobility and need family members’ help in dressing and eating, which are some BADLs. This was confirmed by Kara et al., who concluded that the ADL impairment rate in older adults with chronic diseases was significantly higher than that in those without chronic diseases [41]. Chronic diseases and disabilities continue to be important factors affecting the diminished ADL of middle-aged and older adults. For the prevention of chronic diseases, first, it is important to focus on their initial physical health status and raise awareness of chronic diseases among older adults through health education and free medical check-ups. Second, chronic diseases can be effectively prevented by helping middle-aged and older adults to develop good behavioral habits and participate in social activities, and improved life satisfaction can also help to enhance their health status. Finally, for middle-aged and older people with chronic diseases such as diabetes and hypertension, corresponding health management services should be provided by improving family doctor contracting services [42,43].

## 5. Conclusions

With regard to demographic characteristics, the rate of ADL impairment in older adults increases with age, especially for those older than 80 years of age, which is much greater than for those younger than 80 years of age. This indicates that the risk of ADL impairment increases as physical function declines in older adults [44]. In terms of gender, although the number of women was higher than that of men in the descriptive statistics, the results of the multivariate binary logistic regression analysis showed that ADL impairment was higher among men than women, indicating that men were at greater risk of ADL impairment due to more physical work and family responsibilities in daily life. Although the regression results excluded alcohol consumption, we conducted a stratified analysis of smoking habits, and the regression results demonstrated that older adults who had smoked for a longer period of time and smoked more cigarettes per day on average were more impaired in their ADL than those who had smoked for a shorter period of time and smoked less [45]. In terms of physical health status, older adults with physical disabilities and chronic diseases had much higher ADL impairment than those without physical disabilities and chronic diseases [46], indicating the following: the physical and mental health of older adults urgently requires medical services and care services, more attention should be paid to the health service needs of health disadvantaged groups, and enhancing the cognitive development of older adults and improving their healthy living standards is important.

The contribution of this study is that it utilizes high-quality microdata that can fully reflect the characteristics of aging and dynamically tracks its evolution, which is the basis for systematically conducting research related to aging in the population and has practical significance for improving the impaired ADL of older adults. The influencing factors proposed in this study are a breakthrough. Previous studies have considered ADL in older adults in relation to age, gender, and other factors, but less attention has been paid to smoking as an influential factor. In addition, the present study hopes to help policymakers, researchers, and the public further understand older adults’ situations and assist them in developing better policy plans to enhance living conditions.

This study analyzed only the factors influencing older adults’ ability to perform ADL from the dimension of the degree of ADL impairment, and this has some limitations. Future studies should examine the degree of ADL impairment and its associated factors in a specific group of older adults [47]. Concurrently, the hypothesis of high responsibility attributed to men in this study is limited and should also consider the higher standard of living of women regarding physical activity; hence, the conclusions need to be combined with the specific social context and sample conditions and cannot be generalized. In addition, this is a cross-sectional study. In the future, older adults in different age groups can be followed up on according to long-term follow-up and survey data.

## Data Availability

Data are available in article.

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
