# Peer review of "Impaired Activity of Daily Living Status of the Older Adults and Its Influencing Factors: A Cross-Sectional Study"

_ijerph, 2022, doi:10.3390/ijerph192315607_

Round 1
Reviewer 1 Report
Introduction:
1. Please recheck referencing. It seems like some citations were placed after full stops while others before full stops and some with no space between the last word at the citation number.
2. Please recheck the whole sentence below. The structure of the sentence is hard to follow/understand:
Therefore, studies to understand the factors influencing the ability to perform daily activities in the elderly have found that lifestyle affects the ability to perform daily activities in the elderly, such as smoking, drinking alcohol, engaging in Therefore, we found that the effect of lifestyle on the daily activity ability of the elderly, such as smoking, alcohol consumption, physical work, recreation and exercise, on the daily activity ability of the elderly[5].
3. Please recheck the whole sentence below. The sentence is extremely low with no punctuation. The structure of the sentence is hard to follow/understand:
Thirdly, factor-based studies: different genders, presence of chronic diseases, disabilities and social activities were considered as the main influencing factors on the ability to perform activities of daily living, self-rated health and depression in older adults[14]and the importance of understanding the relationship between activities of daily living (ADL) and depressive symptoms to reduce the incidence of depressive symptoms and improve the quality of life of the elderly[15]used the 2014 Chinese Longitudinal Social Survey on Aging to explore the relationship between activities of daily living, cognitive function, social support, and attitudes toward one's own aging, and developed a structural equation model to test the mediating effect[16]investigated the physical function in relation to activities of daily living (ADLs), taking into account the role of cognitive, psychological, and social factors.
Methods:
1. Please correct mis-spelling in the following sentence:
The survey was conducted in 2011, 2013, and 2013. The survey project was conducted in 150 counties...
Results
1. Descriptive Statistics: this part should be rewritten. Please consult English language assistant to improve the flow and structure of the sentences.
2. Impaired Activities of Daily Living (ADL) - similar comment as above.
Discussion
1. Need to highlight the limitations of the study.
Author Response
Dear Reviewer,
Thank you very much for reviewing the paper and for your high level of professional suggestions. We have carefully read and followed each of your valuable comments on this manuscript, and have invited professional English editors and institutions to make changes to the sentence structure and language of our manuscript, and have added the proofs of touch-ups to the attachment for review. We hope that the changes we have made have further improved the quality of the manuscript, and we will respond to each of your valuable comments as follows.
Point 1:Introduction:
- Please recheck referencing. It seems like some citations were placed after full stops while others before full stops and some with no space between the last word at the citation number.
Response 1:Thank you very much for your meticulous review of this manuscript. Your comments helped us to identify the problems we had with the reference format, so we followed your comments to recheck and revise the citation format of the references to make them more in line with the requirements of the journal and your review work.
Point 2:
- Please recheck the whole sentence below. The structure of the sentence is hard to follow/understand:
Therefore, studies to understand the factors influencing the ability to perform daily activities in the elderly have found that lifestyle affects the ability to perform daily activities in the elderly, such as smoking, drinking alcohol, engaging in Therefore, we found that the effect of lifestyle on the daily activity ability of the elderly, such as smoking, alcohol consumption, physical work, recreation and exercise, on the daily activity ability of the elderly[5].
Response 2:Thank you very much for your comments, which we very much agree with. This was caused by our relatively lack of English language skills, so we consulted a professional English editor to revise and embellish the language throughout the manuscript to improve the quality of the language in this manuscript. Thus the formulation of this paragraph in the article was revised as follows:
(Lines 48-52)Therefore, the study of the impaired ADL of older adults and their influencing factors can help to fully understand the mechanism of health determination and the path of intervention, assist in formulating scientific health interventions, improve the living conditions of older adults, and build a more efficient economic support system for the older people.
Point 3:
- Please recheck the whole sentence below. The sentence is extremely low with no punctuation. The structure of the sentence is hard to follow/understand:
Thirdly, factor-based studies: different genders, presence of chronic diseases, disabilities and social activities were considered as the main influencing factors on the ability to perform activities of daily living, self-rated health and depression in older adults[14]and the importance of understanding the relationship between activities of daily living (ADL) and depressive symptoms to reduce the incidence of depressive symptoms and improve the quality of life of the elderly[15]used the 2014 Chinese Longitudinal Social Survey on Aging to explore the relationship between activities of daily living, cognitive function, social support, and attitudes toward one's own aging, and developed a structural equation model to test the mediating effect[16]investigated the physical function in relation to activities of daily living (ADLs), taking into account the role of cognitive, psychological, and social factors.
Response 3: Thank you very much for your professional opinion, which we very much agree with. We have reviewed this paragraph again and found that it does have the problems you pointed out, such as less punctuation and difficult sentence structure. We have also taken the same approach as the previous problem and invited a professional English editor to revise and review the language of our manuscript to make it smoother and easier to read and understand. The content of this paragraph was revised after the professional editor's touch-ups as follows:
(Lines 69-87) Different genders, the presence of chronic diseases, disabilities, and social activities were considered the main influencing factors on the ability to perform ADLs, self-rated health, and depression in older adults [11]. Understanding the relationship between ADLs and depressive symptoms to reduce the incidence of these symptoms and thus improve the quality of life of older adults [12] has also been researched. Some scholars have explored the relationship between ADLs, cognitive functioning, social support, and their own attitudes toward aging using the 2014 Chinese Longitudinal Survey on Aging Society and developed structural equation models to examine mediating effects. Other scholars have investigated physical functioning related to ADLs, considering the role of cognitive, psychological, and social factors, using physical, cognitive, psychological, and social instruments to investigate the basic activity (BADL) and instrumental activity (IADL) levels [13]. The analyses revealed that physical functioning was the only individual factor significantly associated with BADL and IADL levels in older adults [14]. Some scholars have found that depression status and ADLs have a significant impact on the quality of life of older adults, and believe that comprehensive measures should be taken at community, family, and individual levels to screen for and intervene in depression, focus on the prevention and treatment of chronic diseases, and improve older adults’ quality of life [15,16].
11.Liu, L.; Zhaoyan, Z.; Jiantao, L.; Fangyu, Z.; Shang, W.; Yan, H.. Analysis of activity of daily living ability, health self-assessment and depression status of elderly people in China. Med. Soc. 2020, 33, 90–94.DOI:10.13723/j.yxysh.2020.06.022.
12.Chen, J.F.; Fang, M.W.; ChengHan, X.; Snap, M.. Study on the relationship between activity of daily living ability and depressive symptoms in Chinese elderly people. Chin. Gen. Med. 2020, 23, 2852–2855 + 2862.DOI:10.12114/j.issn.1007-9572.2019.00.693.
13.Wang, G.; Shi, J.; Yao, J.; Fu, H.. Relationship between Activities of Daily Living and attitude toward own aging among the elderly in China: A chain mediating model. Int. J. Aging Hum. Dev. 2020, 91, 581–598. DOI:10.1177/0091415019864595.
14.Candela, F. PhD; Zucchetti, G. PhD; Ortega, E. PhD; Rabaglietti, E. PhD; Magistro, D. PhD. Preventing loss of basic Activities of Daily Living and Instrumental Activities of Daily Living in elderly: Identification of Individual Risk Factors in a Holistic Perspective. Holist. Nurs. Pract. 2015, 29, 313–322. DOI:10.1097/HNP.0000000000000106.
15.Yanhong, L.; Peiling, Z.; Xiuling, S.; Ping, L.; Yaping, Y.; Zhenxiang, Z.. Effects of depression status and activities of daily living ability on quality of life of older adults. Chin. J. Gerontol. 2016, 36, 1179–1181.DOI:CNKI:SUN:ZLXZ.0.2016-05-073.
16.Supawadee Putthinoi;Suchitporn Lersilp;Nopasit Chakpitak.Performance in Daily Living Activities of the Elderly While Living at Home or Being Home-bound in a Thai Suburban Community. Procedia Environ. Sci. 2016, 36.DOI:10.1016/j.proenv.2016.09.015.
Point 4:
Methods:
- Please correct mis-spelling in the following sentence:
The survey was conducted in 2011, 2013, and 2013. The survey project was conducted in 150 counties...
Response 4: Thank you very much for your careful and detailed review and reading, we found the spelling errors in our manuscript, we reworked the sentence and found the spelling errors therein, we are very sorry for the bad reading experience, your comments also made us find the spelling problems in our manuscript, so we have revised the sentence as follows:
(Lines 97-109) The survey was conducted in 150 counties and 450 communities (villages) in 28 provinces. In the summer of 2014, CHARLS conducted a life history survey of all respondents and provided a historical overview of 20,547 respondents out of a sample of 12,250 in the areas of family, marriage, childbirth, employment, education, migration, and health since their birth. In the summer of 2015, the second follow-up interview of the regular survey of the national baseline sample was conducted, and a total of 11,797 and 20,284 myriads were interviewed, with a follow-up success rate of 87%. A total of 412 village households were interviewed regarding their economic history, and 3,797 individuals were interviewed about their oral histories [17].
17.Zhao, Y.; Strauss, J.; Yang, G.; Giles, J.; Hu, P. (P.); Hu, Y.; Lei, X.; Park, A.; Smith, J.P.; Wang, Y.. China health and retirement longitudinal study, 2011–12. National Baseline Users’ Guide; National School of Development, Peking University.2013.
Point 5:
Results
- Descriptive Statistics: this part should be rewritten. Please consult English language assistant to improve the flow and structure of the sentences.
- Impaired Activities of Daily Living (ADL) - similar comment as above.
Response 5: Thank you again for your comments. We have followed your comments completely and invited a professional English assistant to re-touch the section and improve the sentence structure, as well as additions to this section to make the conclusion section richer and better, and our changes are as follows:
(Lines 153-170)
3.1. Descriptive statistic
Among the 10148 older adults ≥60 years old, 4920 were male ( 48.48%), and 5228 were female (51.52%); 5218 people were aged 60–69 (51.42%), 3369 were 70–79 ( 33.20%), and 1561 were ≥80 ( 15.38%); 3865 (38.09%), 5074 (50%), and 1209 people (11.91%) rated their health as good, average, and bad respectively.
3.2. Impaired ADLs
Among the 11 ADL abilities, 7453 older adults had no impairment in ADLs. That is, 73.44% were not impaired in ADLs, and 26.56% were impaired (including 17.34% of mild impairment and 9.22% of severe impairment). Among those impaired, the mild and severe impairment rate was 23.72% and 12.66% for males, and 10.31% and 5.97% for females, respectively. The mild and severe impairment rate was 0.54% and 0.25% for older adults aged 60–69 years, 31.97% and 4.99% for those aged 70–79, and 41.96% and 48.30% for those aged ≥80 years, respectively. By the end of 2015, the number of people over 60 years in China had reached 220 million. After the baseline survey in 2011–12, CHARLS calculated the sample weight according to the sampling procedure. The weighted CHARLS characteristics were often close to those in the 2010 census, indicating that this data fully represented middle-aged people in China [23].
23.Yaohui, Z.; Yafeng, W.; Xinxin, C.;Qinqin, M.; Ye, T.; Tao, Z.; Chao, L.; Dubi, H.; Xueyuan, L.; Xiaomin, Z.; Song, Z.;et al. China Health and Wellness Tracking Survey Project Team, Beijing University. China Health and Wellness Report, 2019.
Point 6:
Discussion
- Need to highlight the limitations of the study.
Response 6: We fully agree with your comments, it is essential for a study to clarify and point out the limitations of the study, identifying the limitations of the study can provide experience and broader research ideas for future studies, therefore we have added the limitations of the study and our modifications are as follows:
(Lines305-314) This study only analyzed the factors influencing the ability to perform ADL in older adults from the dimension of the degree of ADL impairment, which has some limitations. Future studies should be more targeted to examine the degree of ADL impairment and its associated factors in a specific group of older adults[43].Concurrently, the hypothesis of high responsibility attributed to men in this study is limited and should also consider the higher standard of living of women in physical activity, so the conclusions need to be combined with the specific social context and sample conditions and cannot be generalized.In addition, this is a cross-sectional study. In the future, we could follow up with older adults in different age groups and groups according to long-term follow-up and survey data.
43.Trevissón-Redondo;B.;López-López, D.;Pérez-Boal, E.;Marqués-Sánchez, P.;Liébana-Presa, C.;Navarro-Flores, E.;… Becerro-de-Bengoa-Vallejo, R. Use of the Barthel Index to Assess Activities of Daily Living before and after SARS-COVID 19 Infection of Institutionalized Nursing Home Patients. International Journal of Environmental Research and Public Health. 2021,18,7258. DOI:10.3390/ijerph18147258.
Thanks very much for taking your time to review this manuscript. We really appreciate all your generous comments and suggestions, we hope that our revisions will further improve the quality of the manuscript and make your review process smoother.

Reviewer 2 Report
The manuscript under review consists of a survey based on demographic data provided by a periodic population survey in China. The goal is to explore the status of ADL and related factors among the elderly. It seems unclear which specific year the data in the database refers to.
There are some inaccuracies and several weaknesses related to the style of writing, linguistic precision, methodology of the investigation and the arguments produced to discuss the results. The main ones are listed below:
Line 45_drinking alcohol, engaging in Therefore, we found... At least one word is missing
Line 45-48 - the daily activity ability of the elderly, such as smoking, alcohol consumption, physical work, recreation and exercise. Here, risk factors and protective factors are put together without any discussion of their different roles. In fact, it has been widely demonstrated that physical work, recreation and exercise can mediate the effect of negative lifestyles and therefore should be separated.
Line 48-51: Understanding the impact of depressive state ... the authors recommend that prevention measures be taken but that appears to be illogical in the argumentative sequence
Line 62 Second, an experimental study: comparison between self-reported activities of daily living (ADL) and the state of instrumental activities of daily living (IADL) of elderly survivors before ... This paragraph in the intentions of the authors, should present a review of the main ways in which the problem of the ability to perform ADL in the elderly is investigated; however, what has been discussed appears not to be connected to the specific problem. Furthermore, the authors do not justify the choice to report studies on environmental disasters that appear difficult to connect.
Line 79. A targeted screening program is needed to identify ... Corrective interventions are again recommended and this is improper and illogical in this part of the paper
Line 110 -The survey was conducted in 2011, 2013, and 2013. It should be 2015.
The study is carried out on a database of 23,000 subjects produced by a national survey. Of these, 10,148 subjects were selected based on who provided data on their ADL. No precise information is provided to understand how these subjects were extracted, for example: did all the over 60s fill in the ADSL scale or was this optional? The study is carried out on a database of 23,000 subjects produced for a national survey. Of these, 10,148 subjects were selected who provided data on their ADL. No precise information is provided to understand how these subjects were extracted, for example: did all the over 60s fill in the ADL scale or was this optional? What should be the size of the target population? How was the initial sample of 23,000 subjects selected? Furthermore, the initial survey was longitudinal with three collection points: the data in this study from which of these points are they derived?
Descriptivi statstics : 10148 = 60 missing "subjects" in Line 148. The sample description is excessively analytical and describes what is already present in the table: a summary of the main results was enough to characterize the sample; the lack of population size makes it impossible to assess the representativeness of the sample itself.
Among the 11 ADLs, none of the 7453s were damaged ... What does this mean? In the text of the manuscript there are several statements that reveal poor editorial supervision
3.3. Univariate analysis of the activity of the daily life capacity of the 175 older adults: it is once again a mass of results almost never followed by statistical indicators and without a minimum of logical organization
Page 5 -Table 1. Comparison of the activities of the daily life skills among elderly people with different characteristics. The table presents the headings row twice. Table 2. It should be better described as it is ambiguous that binomial variables are on the same row with respect to statistical indicators
Line 214. .... have been considered for smoking as an influencing factor ... In this case they should stratify the smoking habit at least by daily number of cigarettes and by years of exposure to smoke. The presented study divides only between smokers and non-smokers.
Line 285- This was confirmed by the study which concluded that the ADL impairment rate in the elderly with chronic diseases was significantly higher than in those without chronic diseases. In this case the authors use a reference from 1998 when a more recent literature on disability and chronic diseases is just abundant. Furthermore, they state "Therefore, it is very necessary to actively carry out the prevention and control of chronic diseases in the elderly ..." health education and free medical examinations, so that they can be treated promptly in case of illness ... However, if the disease is chronic, what can be prevented are only the consequences. In fact, the critical link with the objective of the study seems to be weak in the conclusions; the relationship between "habits and chronic disease" (causes) and the "ADL reduction" (consequences) in old age should be traced back to preventable causes.
Author Response
Dear Reviewer,
Thanks very much for taking your time to review this manuscript. We really appreciate all your generous comments and suggestions.We have revised the content of the manuscript in accordance with your comments, and have also turned to a professional English editor to touch up the language of the manuscript. Here are the changes we have made in response to the scientific corrections you suggested.
Point 1:The manuscript under review consists of a survey based on demographic data provided by a periodic population survey in China. The goal is to explore the status of ADL and related factors among the elderly. It seems unclear which specific year the data in the database refers to.
Response 1:Thank you very much for pointing out the problem, it was our mistake that we failed to specify the year of the database data in the manuscript, so we have clarified it in the manuscript, and we also respond to your comments here.
The database used in this study is the China Health and Recuperation Tracking Survey (CHARLS), which aims to provide basic data for studying the issue of aging in the population. This study is based on the second follow-up survey data of the 2015 National Baseline Survey and combines the baseline data of 2011 and the first follow-up survey data of 2013. The survey design was rigorous and standardized based on international experience and the actual situation in China to ensure the scientific nature and representativeness of the sample.
Point 2:There are some inaccuracies and several weaknesses related to the style of writing, linguistic precision, methodology of the investigation and the arguments produced to discuss the results. The main ones are listed below:
Line 45_drinking alcohol, engaging in Therefore, we found... At least one word is missing
Line 45-48 - the daily activity ability of the elderly, such as smoking, alcohol consumption, physical work, recreation and exercise. Here, risk factors and protective factors are put together without any discussion of their different roles. In fact, it has been widely demonstrated that physical work, recreation and exercise can mediate the effect of negative lifestyles and therefore should be separated.
Line 48-51: Understanding the impact of depressive state ... the authors recommend that prevention measures be taken but that appears to be illogical in the argumentative sequence
Response 2:We are extremely grateful to Reviewer for pointing out this problem.We strongly agree with you that risk factors and protective factors should indeed be discussed separately in terms of their different roles for a more scientific presentation, so we have reworked the discussion in this section and adjusted the cited literature to make the content more logical. Our revisions are as follows.:
(Lines45-52) Older adults who smoke and drink too much alcohol have a low capacity for ADLs; engaging in appropriate physical work, recreational activities, and exercise can moderate the effects of negative lifestyles, resulting in a higher capacity [5,6]. Therefore, the study of the impaired ADL of older adults and their influencing factors can help to fully understand the mechanism of health determination and the path of intervention, assist in formulating scientific health interventions, improve the living conditions of older adults, and build a more efficient economic support system for the older people.
- Ricci, Natalia Aquaroni; Francisco, Cristina Oliveira; Rebelatto, Marcelo Nascimento; Rebelatto, José Rubens. Influence of history of smoking on the physical capacity of older people. Arch. Gerontol. Geriatr. 2011, 52, 79–83. DOI:10.1016/j.archger.2010.02.004.
- Wu, B.Y.; Li, J.L.; Liu, W.H.; Wang, Y.Y.; Wang, Z.X. Influence of lifestyle on daily activity capacity of the elderly. China Public Health. 2019, 35, 881–884.DOI:10.11847/zgggws1116087.
Point 3: Line 62 Second, an experimental study: comparison between self-reported activities of daily living (ADL) and the state of instrumental activities of daily living (IADL) of elderly survivors before ... This paragraph in the intentions of the authors, should present a review of the main ways in which the problem of the ability to perform ADL in the elderly is investigated; however, what has been discussed appears not to be connected to the specific problem. Furthermore, the authors do not justify the choice to report studies on environmental disasters that appear difficult to connect.
Line 79. A targeted screening program is needed to identify ... Corrective interventions are again recommended and this is improper and illogical in this part of the paper
Response 3: Thank you very much for your valuable comments, which we strongly agree with. The discussion in this section is indeed lacking as well as less relevant to the research topic we want to explore in this paper, so we have re-read the relevant literature and made a lot of changes and adjustments to these two sections, and our main changes are as follows:
(Lines53-88) Scholars have studied the impaired ADLs of older adults and their influencing factors in two main ways: Firstly, they have used different research methods. For example, some scholars have conducted a comprehensive assessment of older adults over 65 and analyzed the relationship between muscle strength, cognitive function, ADLs, and depression variables in an experiment in which statistical significance was found in all variables [7]. Others have analyzed the mediating effect of ADL ability between diabetes and depression in older adults using difference tests, stepwise regression analysis, and bootstrap mediation tests [8]. A non-randomized controlled intervention trial was also used to divide older adults into a rehabilitation group and a non-rehabilitation group and to measure their ability to perform ADLs through continuous observation and assessment of the quality of life using the Philadelphia Geriatric Center Psychiatric Inventory [9]. Based on relevant literature, most studies on older adults' ADL ability have used statistical analysis of quantitative studies as a tool, relatively mature ADL scales for scoring, and aimed to produce statistically significant results through regression analysis and correlation analysis [10].
Secondly, they have studied the factors influencing the ability of older adults to perform ADLs. Different genders, the presence of chronic diseases, disabilities, and social activities were considered the main influencing factors on the ability to perform ADLs, self-rated health, and depression in older adults [11]. Understanding the relationship between ADLs and depressive symptoms to reduce the incidence of these symptoms and thus improve the quality of life of older adults [12] has also been researched. Some scholars have explored the relationship between ADLs, cognitive functioning, social support, and their own attitudes toward aging using the 2014 Chinese Longitudinal Survey on Aging Society and developed structural equation models to examine mediating effects. Other scholars have investigated physical functioning related to ADLs, considering the role of cognitive, psychological, and social factors, using physical, cognitive, psychological, and social instruments to investigate the basic activity (BADL) and instrumental activity (IADL) levels [13]. The analyses revealed that physical functioning was the only individual factor significantly associated with BADL and IADL levels in older adults [14]. Some scholars have found that depression status and ADLs have a significant impact on the quality of life of older adults, and believe that comprehensive measures should be taken at community, family, and individual levels to screen for and intervene in depression, focus on the prevention and treatment of chronic diseases, and improve older adults’ quality of life [15,16].
7.Won, C.J.; Sangbeom, K.; Jung, H.-Y.. The Casual Relationship among Muscular strength, Cognitive function, activities of daily living, Depression of the elderly. J. Converg. Inf. Technol. 2021, 11, 242–250.DOI:10.22156/CS4SMB.2021.11.05.242.
8.Mei, Y.. Wang Wanchen;Liu Xinlu;Li Qiusha;Fan Chengxin;Song Jia;Wang Xiangyin;Yin Wenqiang. Study on the mediating effect of activity of daily living ability between diabetes and depression in the elderly. Mod. Prev. Med. 2022, 49, 3553–3557 + 3578. DOI:10.20043/j.cnki.MPM.202203329.
9.Imanishi, Miyuki; Tomohisa, Hisao; Higaki, Kazuo. Impact of continuous in-home rehabilitation on quality of life and activities of daily living in elderly clients over 1 year. Geriatr. Gerontol. Int. 2017, 17, 1866–1872. DOI:10.1111/ggi.12978.
10.Kadar, Masne; Ibrahim, Suhaili; Razaob, Nor Afifi; Chai, Siaw Chui; Harun, Dzalani. Validity and reliability of a Malay version of the Lawton instrumental activities of daily living scale among the Malay speaking elderly in Malaysia. Aust. Occup. Ther. J. 2018, 65, 63–68. DOI:10.1111/1440-1630.12441.
11.Liu, L.; Zhaoyan, Z.; Jiantao, L.; Fangyu, Z.; Shang, W.; Yan, H.. Analysis of activity of daily living ability, health self-assessment and depression status of elderly people in China. Med. Soc. 2020, 33, 90–94.DOI:10.13723/j.yxysh.2020.06.022.
12.Chen, J.F.; Fang, M.W.; ChengHan, X.; Snap, M.. Study on the relationship between activity of daily living ability and depressive symptoms in Chinese elderly people. Chin. Gen. Med. 2020, 23, 2852–2855 + 2862.DOI:10.12114/j.issn.1007-9572.2019.00.693.
13.Wang, G.; Shi, J.; Yao, J.; Fu, H.. Relationship between Activities of Daily Living and attitude toward own aging among the elderly in China: A chain mediating model. Int. J. Aging Hum. Dev. 2020, 91, 581–598. DOI:10.1177/0091415019864595.
14.Candela, F. PhD; Zucchetti, G. PhD; Ortega, E. PhD; Rabaglietti, E. PhD; Magistro, D. PhD. Preventing loss of basic Activities of Daily Living and Instrumental Activities of Daily Living in elderly: Identification of Individual Risk Factors in a Holistic Perspective. Holist. Nurs. Pract. 2015, 29, 313–322. DOI:10.1097/HNP.0000000000000106.
15.Yanhong, L.; Peiling, Z.; Xiuling, S.; Ping, L.; Yaping, Y.; Zhenxiang, Z.. Effects of depression status and activities of daily living ability on quality of life of older adults. Chin. J. Gerontol. 2016, 36, 1179–1181.DOI:CNKI:SUN:ZLXZ.0.2016-05-073.
16.Supawadee Putthinoi;Suchitporn Lersilp;Nopasit Chakpitak.Performance in Daily Living Activities of the Elderly While Living at Home or Being Home-bound in a Thai Suburban Community. Procedia Environ. Sci. 2016, 36.DOI:10.1016/j.proenv.2016.09.015.
Point 4:Line 110 -The survey was conducted in 2011, 2013, and 2013. It should be 2015.
Response 4: Thank you very much for your careful review and pointing out the problems we have, this is our mistake, so after getting your review comments we have revised this statement.
Point 5: The study is carried out on a database of 23,000 subjects produced by a national survey. Of these, 10,148 subjects were selected based on who provided data on their ADL. No precise information is provided to understand how these subjects were extracted, for example: did all the over 60s fill in the ADSL scale or was this optional? The study is carried out on a database of 23,000 subjects produced for a national survey. Of these, 10,148 subjects were selected who provided data on their ADL. No precise information is provided to understand how these subjects were extracted, for example: did all the over 60s fill in the ADL scale or was this optional? What should be the size of the target population? How was the initial sample of 23,000 subjects selected? Furthermore, the initial survey was longitudinal with three collection points: the data in this study from which of these points are they derived?
Response 5: Thank you very much for your scientific review of the manuscript, which gave us the opportunity to further reflect on our shortcomings and the space to further improve the quality of the manuscript's content. In response to your questions, our answers are as follows:
In the database used for this study, we found that not all older adults over 60 years of age responded to the ADSL scale, and responses to this question were selective; therefore, in the initial data screening we excluded older adults who did not respond to the scale and screened out the sample that answered the scale completely, resulting in 10,148 valid sample data.
The data for this study were obtained from the baseline survey of the 2015 China Health and Aging Tracking Survey (CHATS), which was conducted in 150 counties and 450 communities (villages) in 28 provinces across China. In the summer of 2014, CHARLS conducted a special survey on the life history of all respondents, describing the historical panorama of 20,547 respondents out of a sample of 12,250, in terms of family, marriage, parenting, employment, education, migration, and health since their birth, with a follow-up success rate of 86%. In the summer of 2016, we conducted a survey on the oral history of people over 80 years of age and a survey on the economic history of grassroots people in the early republic in a sample of village houses, and completed a total of 412 interviews on the economic history of village houses and 3,797 interviews on the oral history of people.
This study is based on the second follow-up interview of this survey and the supplementary survey in 2016, with a total of about 23,000 samples. Also, according to the need of the purpose of this study, elderly people aged 60 and above were selected as the study population, and missing and outlier samples were screened out, and respondents who had answered questions related to the ADL scale were selected, and finally a total of 10,148 samples were included in this study.
Point 6:Descriptivi statstics : 10148 = 60 missing "subjects" in Line 148. The sample description is excessively analytical and describes what is already present in the table: a summary of the main results was enough to characterize the sample; the lack of population size makes it impossible to assess the representativeness of the sample itself.
Response 6: Thank you very much for your valuable comments, and we agree that the lack of sample size does make it difficult to assess its representativeness, so we have reviewed relevant information to provide additional clarification for this study.
By the end of 2015, there were 220 million people over 60 years of age in China. 2011-12 After the baseline survey, CHARLS calculated the sample weights according to the sampling procedure, and the weighted CHARLS demographic characteristics are very close to those of the 2010 Population Census, indicating that the data are very representative of the middle-aged population in China. In terms of the sampling method, CHARLS adopts a strict random sampling, and all county-level units in the country are randomly selected according to the PPS (Probability Proportional to Size) method stratified by region, urban and rural areas, and bogus GDP. At the same time, in each stage of the sampling process, in order to avoid artificial manipulation, the sampling was conducted by the project staff using a computer program, and no substitution of samples was allowed. CHARLS successfully conducted regular follow-up visits to these baseline samples in 2013, 2014 and 2015, based on the 2011-12 national large-scale baseline survey, to maintain the representativeness of the middle-aged population.
Point 7:Among the 11 ADLs, none of the 7453s were damaged ... What does this mean? In the text of the manuscript there are several statements that reveal poor editorial supervision
Response 7: Thank you very much for your careful review, which helped us to identify the linguistic deficiencies of the manuscript. We have therefore consulted a professional English editor to rework and retouch the entire manuscript language to improve the quality of the manuscript language, and we have included the retouching certificate in the attached document for your convenience.
Point 8: 3.3. Univariate analysis of the activity of the daily life capacity of the 175 older adults: it is once again a mass of results almost never followed by statistical indicators and without a minimum of logical organization
Response 8: Your comments are very important and scientific. This is a shortcoming in our exposition, where we only described the data repeatedly and did not present statistical indicators. Therefore, we have reviewed the relevant literature and carefully studied its presentation during factor analysis, while making the following changes for our study:
(lines171-187) Comparing the ADL ability of older adults in different populations, the differences in gender, age, marital status, education, exercise, smoking status, alcohol consumption, whether disabled, whether suffering from chronic diseases, and self-rated health were statistically significant (P<0.01).
To determine whether there is multicollinearity in a data set, we must look at the absolute value of the correlation coefficient between two independent variables; usually, if the value is greater than 0.75, it is considered that there is a linear relationship between the two variables. The results of the two-factor correlation analysis of all explanatory variables in the database showed that the correlation coefficients between the variables were less than 0.75, so there was no multicollinearity between the variables. After the Monte Carlo test was selected for the chi-square test data statistics, the conclusion reached was consistent with the Pearson test. (Tables 2–3)
Point 9: Page 5 -Table 1. Comparison of the activities of the daily life skills among elderly people with different characteristics. The table presents the headings row twice. Table 2. It should be better described as it is ambiguous that binomial variables are on the same row with respect to statistical indicators
Response 9: Thank you very much for your suggestions to improve the quality of the table in our manuscript, we fully agree and have made changes according to your comments: in Table 1 we have removed the recurring table headers; in Table 2 we have arranged the control groups of the binary variables vertically by studying the literature in the related fields and combining your modifications. Once again, thank you for your experienced modifications, and the adjustments made according to your comments made the data presentation clearer.
Point 10: Line 214. .... have been considered for smoking as an influencing factor ... In this case they should stratify the smoking habit at least by daily number of cigarettes and by years of exposure to smoke. The presented study divides only between smokers and non-smokers.
Response 10: Thank you very much for your comments on the modification of smoking as an influencing factor, and we strongly agree with you after careful consideration. Therefore, we found the original data for analysis and performed a stratified and multifactorial binary logistic regression analysis of smoking factors according to years of exposure to smoke (less than 20 years, 20-40 years, and more than 40 years) and the average number of cigarettes smoked per day (less than 20, 20-40, and more than 40 cigarettes) among older adults, which showed that years of smoking in the range of 20-40 years (P< 0.01), the average number of cigarettes smoked per day was 20-40 (P<0.01) and more than 40 (P<0.05), indicating that the years of smoking (20-40 years) and the number of cigarettes smoked per day (>20) had a significant effect on the impairment of activities of daily living in older adults.
Point 11:Line 285- This was confirmed by the study which concluded that the ADL impairment rate in the elderly with chronic diseases was significantly higher than in those without chronic diseases. In this case the authors use a reference from 1998 when a more recent literature on disability and chronic diseases is just abundant. Furthermore, they state "Therefore, it is very necessary to actively carry out the prevention and control of chronic diseases in the elderly ..." health education and free medical examinations, so that they can be treated promptly in case of illness ... However, if the disease is chronic, what can be prevented are only the consequences. In fact, the critical link with the objective of the study seems to be weak in the conclusions; the relationship between "habits and chronic disease" (causes) and the "ADL reduction" (consequences) in old age should be traced back to preventable causes.
Response 11: Thank you very much for your comments, which we strongly agree with. We have therefore followed your comments to update the references, and in the process of updating the references, we have also reorganized the logic of the section in the hope of improving the weaknesses of our exposition and increasing the link between the conclusions and the research objectives, and our modifications are as follows:
(lines264-285) Disability and chronic diseases are important factors affecting older adults' ability to perform ADLs. Those with disabilities are less able to take care of themselves, and older adults with chronic diseases are limited in their ability to exercise to a certain extent, and their ability to express themselves is also greatly reduced [37,38]. Some scholars have specifically proposed to develop a scoring system based on a ratio scale for measuring disability in personal and social ADLs to establish the degree of disability in older adults [39]. Older adults with chronic diseases have reduced physical functions, weak resistance, and poor mobility, and they need family members' help in dressing and eating, which are some BADLs. This was confirmed by the study, which concluded that the ADL impairment rate in older adults with chronic diseases was significantly higher than that in those without [40]. Chronic diseases and disabilities continue to be important factors affecting the diminished ADL of middle-aged and older adults. For the prevention of chronic diseases, firstly, it is important to focus on their initial physical health status and raise awareness of chronic diseases among older adults through health education and free medical check-ups. Secondly, chronic diseases can be effectively prevented by helping middle-aged and older adults to develop good behavioral habits and participate in social activities, and improved life satisfaction can also help to enhance their health status. Finally, for middle-aged and older people already suffering from chronic diseases, corresponding health management services should be provided for those with diseases such as diabetes and hypertension by improving family doctor contracting services [41,42].
37.Jie, Jin-Hua; Li, Dan; Jia, Li-Na; Chen, Yifeng; Yang, Yan; Zheng, Bailing; Wu, Chuancheng; Liu, Baoying; Xu, Rongxian; Xiang, Jianjun; et al. Activities of daily living and its influencing factors for older people with type 2 diabetes mellitus in urban communities of Fuzhou, China. Front. Public Health. 2022, 10, 948533. DOI:10.3389/fpubh.2022.948533.
38.Karakurt, Papatya; Ünsal, Ayla. Fatigue, anxiety and depression levels, activities of daily living of patients with chronic obstructive pulmonary disease. Int. J. Nurs. Pract. 2013, 19, 221–231. DOI:10.1111/ijn.12055.
39.Liu Y; Chen LP; Gao E; et al. Effects of smoking on lung function in middle-aged and elderly people. Chinese Journal of Gerontology. 2013,33,4247-4248. DOI:10.3969/j.issn.1005-9202.2013.17.068.
40.Ćwirlej-SozaÅ„ska, Agnieszka Beata; SozaÅ„ski, Bernard; WiÅ›niowska-Szurlej, Agnieszka; Wilmowska-PietruszyÅ„ska, Anna A,et al. An assessment of factors related to disability in ADL and IADL in elderly inhabitants of rural areas of south-eastern Poland. Ann. Agric. Environ. Med. 2018, 25, 504–511. DOI:10.26444/aaem/81311.
41.Kara, Osman; Soysal, Pinar; Kiskac, Muharrem; Smith, Lee; Karışmaz, Abdülkadir; Kazancioglu, Rumeyza. Investigation of optimum hemoglobin levels in older patients with chronic kidney disease. Aging Clin. Exp. Res. 2022. DOI:10.1007/s40520-022-02246-1.
42.Tang, Q.; Yuan, Min; Wu, Wenhui; Wu, Huanyun; Wang, Cao; Chen, Gang; Li, Chengyue; Lu, Jun. Health status and individual care needs of disabled elderly at home in different types of care. Int. J. Environ. Res. Public Health. 2022, 19, 1371. DOI:10.3390/ijerph191811371.
These are our main revisions, and we thank you again for your extensive work in reviewing this manuscript and for your scientific comments, which gave us the opportunity to reflect on our problems and to make changes and improve the quality of the manuscript.Thank you and all the reviewers for the kind advice again.

Reviewer 3 Report
The paper is very interesting expecially for the potential social impact. The gender difference found are the major point of interpretation. The hypothesis to attribute this behaviour to the higer responsability of the male gender is restricted. Some considerazions of the higer livel of the dalla phyical activity of the women should be considered. The authors need to specify that the conclusions canot be shared in other context.
Author Response
Point 1:The paper is very interesting expecially for the potential social impact. The gender difference found are the major point of interpretation. The hypothesis to attribute this behaviour to the higer responsability of the male gender is restricted. Some considerazions of the higer livel of the dalla phyical activity of the women should be considered. The authors need to specify that the conclusions canot be shared in other context.
Response1 :
Dear Reviewer,
Thank you very much for reviewing this manuscript and for your kind comments. Your comments were very valuable in improving the quality of our manuscript, and we have carefully followed your comments and improved the content of the manuscript and consulted with a professional English editor to improve the entire manuscript. based on the database utilized for this study and that we should take into account the importance of women in daily physical activity and not only attribute it to the assumption of high responsibility of men, and state in the manuscript that these findings cannot be shared in other contexts. In addition, we have refined and improved the data in the article, strengthened the thesis section, and updated and formatted the references to make the manuscript more compliant with the review requirements.
Your valuable comments have been very helpful to our study and have allowed us to justify and refine our findings in a more rigorous and scientific manner.We would like to thank the referee again for taking the time to review our manuscript.

Round 2
Reviewer 1 Report
INTRODUCTION:
1. Suggestion:
Consider changing the following:
Older adults who smoke and drink too much alcohol have a low capacity for ADLs; engaging in appropriate physical work, recreational activities, and exercise can moderate the effects of negative lifestyles, resulting in a higher capacity [5,6].
To:
Older adults who smoke and drink too much alcohol have a low capacity for ADLs; while engagement in physical work, recreational activities, and exercise can moderate the effects of negative lifestyles, resulting in a higher capacity [5,6].
2. Suggestion:
To remove the following from the sentence:
"... and aimed to produce statistically significant results through regression analysis and 66 correlation analysis".
3. Suggestion:
Consider changing the following:
Understanding the relationship between ADLs and depressive symptoms to reduce the incidence of these symptoms and thus improve the quality of life of older adults [12] has also been researched.
To:
Research to understand the relationship between ADLs and depressive symptoms [12] has been undertaken.
4. Comment:
The following sentence is still hard to follow and confusing:
"...believe that comprehensive measures should be taken at community, family, and individual levels to screen for and intervene in depression, focus on the prevention and treatment of chronic 85 diseases, and improve older adults’ quality of life [15,16]."
Consider rephrasing or simplifying the sentence.
METHODS:
1. Comment:
Please see the following sentence:
"...and 3,797 individuals were interviewed about their oral histories."
What is "oral histories"?
2. Comment:
Consider changing the following:
(1) Independent variables: Demographic characteristics [19], lifestyle habits [20], health status [21], and other relevant factors that may affect the ability of older adults to perform ADLs were included, incorporating gender, age, marital status, education level, exercise, smoking, drinking, disability, having chronic diseases, and self-rated health status, to examine the effects on the ADL ability of older adults [22]
To:
Relevant factors that may affect the ability of older adults to perform ADLs were included as independent variables (e.g., demographic characteristics [19], lifestyle habits [20], health status [21], and other relevant factors). Data on gender, age, marital status, education level, exercise, smoking, drinking, disability, chronic diseases, and self-rated health status were used in the analysis to examine the effects on the ADL ability of older adults [22].
RESULTS:
1. Descriptive statistic
Please do not write all of them in one sentence. Separate relevant points in another sentence.
2. Comment:
Sub-section 3.1.
The title should be just "ADL status"
3. Comment:
I think there is no need to explain in detail about multicollinearity in the results section.
My suggestion is to just state simply that there was no multicollinearity between the variables.
Table 2. Multicollinearity test (Pearson correlation analysis) can be placed in appendix.
4. Comment
I still see some spelling mistakes, grammatical errors, and punctuation issues in this section. For exampled:
"...per day (20-40 (P<0.01), and 40 Above (P<0.05)..."
"...was consistent with the Pearson test. (Tables 2–3)"
DISCUSSION
1. Comment:
"This study showed that the ADL impairment rate of Chinese older adults was 26.56%..."
Rather than referring to Chinese older adults, it is better to refer them to older adults in the CHARLS sample.
2. Comment:
See the following sentence:
"Thus, they risk physical injuries, and..."
Do you mean "they are more at risk"?
CONCLUSION
1. Comment:
Shift the strength of the study to the discussion section.
Focus the content of the conclusion on what has been found in the study.
OTHER COMMENTS
It seems to me that the writing of the discussion section can be further improved with another round of professional editing.
Author Response
Dear Reviewers, Thank you very much for your review and revision of this manuscript. Your suggestions were very professional and scientific, and at the same time very valuable to us. We feel your seriousness and responsibility in the review process and express our sincere gratitude to you.The main corrections in the paper and the responds to your comments are as flowing:
Point 1:
Consider changing the following:
Older adults who smoke and drink too much alcohol have a low capacity for ADLs; engaging in appropriate physical work, recreational activities, and exercise can moderate the effects of negative lifestyles, resulting in a higher capacity [5,6].
To:
Older adults who smoke and drink too much alcohol have a low capacity for ADLs; while engagement in physical work, recreational activities, and exercise can moderate the effects of negative lifestyles, resulting in a higher capacity [5,6].
Consider changing the following:
Understanding the relationship between ADLs and depressive symptoms to reduce the incidence of these symptoms and thus improve the quality of life of older adults [12] has also been researched.
To:
Research to understand the relationship between ADLs and depressive symptoms [12] has been undertaken.
Consider changing the following:
(1) Independent variables: Demographic characteristics [19], lifestyle habits [20], health status [21], and other relevant factors that may affect the ability of older adults to perform ADLs were included, incorporating gender, age, marital status, education level, exercise, smoking, drinking, disability, having chronic diseases, and self-rated health status, to examine the effects on the ADL ability of older adults [22]
To:
Relevant factors that may affect the ability of older adults to perform ADLs were included as independent variables (e.g., demographic characteristics [19], lifestyle habits [20], health status [21], and other relevant factors). Data on gender, age, marital status, education level, exercise, smoking, drinking, disability, chronic diseases, and self-rated health status were used in the analysis to examine the effects on the ADL ability of older adults [22].
Response 1:
Dear reviewers, thank you very much for your comments on the language of this manuscript, your corrections have significantly improved the language quality of our manuscript and made up for the linguistic and grammatical deficiencies of this manuscript, we think your corrections are very scientific and professional, so we have summarized the language corrections you made to this manuscript, and we have carefully followed your comments to replace the sentence expressions in the manuscript. Thank you again for your comments and for giving us the opportunity to learn and improve the quality of the manuscript. We have marked the changes made in the text exactly as you suggested for your review.
Point 2:
To remove the following from the sentence:
"... and aimed to produce statistically significant results through regression analysis and 66 correlation analysis".
Response 2:
Dear Reviewer, Thank you very much for your comments. As you suggested, we should remove the lengthy explanatory section from the sentences. The results of the scholarly research need to be presented more simply and clearly to make the language clearer and more logical, so we have removed this section from the sentences exactly as you suggested, and the changes we have made are marked in the text to facilitate your review.
Point 3:
The following sentence is still hard to follow and confusing:
"...believe that comprehensive measures should be taken at community, family, and individual levels to screen for and intervene in depression, focus on the prevention and treatment of chronic 85 diseases, and improve older adults’ quality of life [15,16]."
Consider rephrasing or simplifying the sentence.
Response 3:
Dear reviewers, we are very sorry that our mistake made the language in this section not smooth to understand, your suggestion made us fully aware of this problem, and we fully agree with your professional opinion. Therefore, we have reorganized and revised the sentences in this section, and our changes are as follows: (also marked in the text)
(lines 80-86): Some scholars have found that depression has a significant impact on the ability of the elderly to perform activities of daily living, and the higher the level of depression, the more severely the elderly are impaired in activities of daily living [15]. Therefore, communities and families should take measures to screen and intervene in the mental health of the elderly in a timely manner to improve the ability of the elderly to perform activities of daily living. [16].
Point 4:
Please see the following sentence:
"...and 3,797 individuals were interviewed about their oral histories."
What is "oral histories"?
Response 4:
Dear Reviewers, We are sorry that our mistake made the presentation of this section hard to understand. We have reviewed the data and found that the 3,797 elderly people are not in particularly good physical condition to write by hand, so they have taken the verbal approach to present their family history of change, reflecting the changes in society from the side. For ease of comprehension, we have modified the presentation of this section as follows:(also marked in the text)
(line 107): 3,797 individuals were interviewed about their family histories.
Point 5:
Descriptive statistic
Please do not write all of them in one sentence. Separate relevant points in another sentence.
Sub-section 3.1.
The title should be just "ADL status"
Response 5:
Dear Reviewer, Thank you very much for your scientific and professional comments. We agree that the description of this section is more in line with your proposed title "ADL status" and that the presentation of the current status is not suitable for a single sentence. Therefore, we have fully followed your comments to improve the presentation of this section, and we have made the following changes: (also marked in the text)
(lines 153-164) 3.1. ADL status
Among 10,148 elderly people aged ≥60 years, 5,218 (51.42%) were aged 60-69 years, 3,369 (33.20%) were aged 70-79 years, and 1,561 (15.38%) were aged ≥80 years. The gender was male 4920 (48.48%) and female 5228 (51.52%). Also in the sample of elderly people who smoked were 5395 (53.16%), 4753 (46.84%) who did not smoke; 2841 (28.00%) who had physical disabilities, 7307 (72.00%) who did not have disabilities; 2908 (32.37%) who had chronic diseases, 7240 (67.63%) who did not have chronic diseases. In the evaluation of their own health, 3865 people (38.09%) rated themselves as having good health, 5074 people (50%) rated themselves as having average health, and 1209 people (11.91%) rated themselves as having bad health. In addition, statistics were also made on the marital status, education level, participation in exercise and alcohol consumption of the elderly.
Point 6:
I think there is no need to explain in detail about multicollinearity in the results section.
My suggestion is to just state simply that there was no multicollinearity between the variables.
Table 2. Multicollinearity test (Pearson correlation analysis) can be placed in appendix.
Response 6:
Dear Reviewer, Thank you again for your professional comments. We strongly agree with your comments and have reviewed the relevant materials and papers, and found that the analysis of multiple covariance does not need to occupy a large amount of content to justify, so we have followed your comments to simplify the presentation of this section by simply stating that the data do not have the problem of multiple covariance, and we have also included Table 2 as an appendix that does not appear in the text. Thank you again for your scientific comments, which made this study more concise. Our modifications are as follows.
(lines 186-191) The results of the two-factor correlation analysis of all explanatory variables in the database showed that the correlation coefficients between the variables were less than 0.75, so there was no multicollinearity between the variables. After the Monte Carlo test was selected for the chi-square test data statistics, the conclusion reached was consistent with the results of Pearson’s correlation analysis.
Point 7:
I still see some spelling mistakes, grammatical errors, and punctuation issues in this section. For exampled:
"...per day (20-40 (P<0.01), and 40 Above (P<0.05)..."
"...was consistent with the Pearson test. (Tables 2–3)"
Response 7:
Dear Reviewer, Thank you again for your careful review of this manuscript. We apologize for the spelling, grammar and punctuation errors in the manuscript due to our mistakes. We have again consulted with the English editor for a second touch-up of the manuscript, and the errors have been corrected to make the manuscript more fluent. The changes we have made are as follows: (marked in the text)
(lines 189-191)After the Monte Carlo test was selected for the chi-square test data statistics, the conclusion reached was consistent with the results of Pearson’s correlation analysis.
(lines 205-210)The results showed that the years of smoking (P<0.01) and average number of cigarettes smoked per day (20–40 [P<0.01] and above 40 [P<0.05]) were associated with a higher rate of impaired ADL in older adults, indicating that the number of years of smoking (20–40) and the number of cigarettes smoked (>20) had a significant effect on older adults’ impaired ADL.
Point 8:
"This study showed that the ADL impairment rate of Chinese older adults was 26.56%..."
Rather than referring to Chinese older adults, it is better to refer them to older adults in the CHARLS sample.
Response 8 :
Dear Reviewer, Thank you very much for your professional comments. We agree with your comments and fully comply with them by changing "Chinese older adults" to "older adults in the CHARLS sample" to make the language more scientific and accurate. We have marked the changes in the text to make it easier for you to review.
Point 9:
See the following sentence:
"Thus, they risk physical injuries, and..."
Do you mean "they are more at risk"?
Response 9 :
Dear Reviewer, We apologize for the difficulties we caused in your understanding due to our inaccurate presentation. We wanted to convey that men may be more at risk in their lives, as you understood, so we have followed your comments and revised this section to read: "they are more at risk". Thank you again for your careful scientific review and revision of this manuscript.
Point 10:
Shift the strength of the study to the discussion section.
Focus the content of the conclusion on what has been found in the study.
Response10 :
Dear Reviewers, Thank you very much for pointing out the weaknesses in the exposition of this manuscript, and the conclusion and discussion sections do need further revision and improvement. We have therefore added to the literature and improved the discussion in the discussion section, and enhanced the focus on the findings themselves in the conclusion section,For the discussion and conclusion sections we have made complete revisions and linguistic touches that we have not listed in our response letter, but have highlighted in yellow in the text for your review. (Lines 215-322)
Point 11:
It seems to me that the writing of the discussion section can be further improved with another round of professional editing.
Response11 :
Dear Reviewer, Thank you very much for your comments. We fully agree with your comments and have consulted with a professional English assistant to make secondary corrections to the discussion section and to add and improve the content of the manuscript in conjunction with your previous comment.
These are the main changes we have made to the manuscript. We hope that the changes we have made will further improve the quality of the manuscript and at the same time make the review process smoother, and once again, we would like to express our sincere gratitude to you.

Reviewer 2 Report
Much improved, the authors seem to have addressed the weaknesses
Author Response
Dear reviewer,
Thank you very much for your approval of this manuscript. We were pleased to know that our work was rated as potentially acceptable for publication in Journal, subject to adequate revision. We thanks for the time and effort that you have put into reviewing the previous version of the manuscript. Your suggestions have enabled us to improve our work. Based on the instructions provided in your letter, we uploaded the file of the revised manuscript.We tried our best to improve the manuscript and made some changes in the manuscript. These changes will not influence the content and framework of the paper. And here we did not list the changes but marked in yellow in revised paper.
We appreciate for your warm work earnestly, and thank you again for your scientific comments and approval of this manuscript.
